# Comparison of Medical and Surgical Treatment in Severe Bell’s Palsy

**DOI:** 10.3390/jcm11030888

**Published:** 2022-02-08

**Authors:** Yong Kim, Seung Geun Yeo, Hwa Sung Rim, Jongha Lee, Dokyoung Kim, Sung Soo Kim, Dong Choon Park, Jae Yong Byun, Sang Hoon Kim

**Affiliations:** 1Department of Rehabilitation Medicine, College of Medicine, Kyung Hee University, Seoul 02447, Korea; okkeun@nate.com (Y.K.); lukaslee@hanmail.net (J.L.); 2Department of Otorhinolaryngology, Head and Neck Surgery, College of Medicine, Kyung Hee University, Seoul 02447, Korea; marslover@naver.com (H.S.R.); otorhino512@naver.com (J.Y.B.); hoon0700@naver.com (S.H.K.); 3Department of Anatomy and Neurobiology, College of Medicine, Kyung Hee University, Seoul 02447, Korea; dkim@khu.ac.kr; 4Medical Research Center for Bioreaction to Reactive Oxygen Species and Biomedical Science Institute, Graduate School, College of Medicine, Kyung Hee University, Seoul 02447, Korea; sgskim@khu.ac.kr; 5Department of Obstetrics and Gynecology, St. Vincent’s Hospital, The Catholic University of Korea, Suwon 16247, Korea; park.dongchoon@gmail.com

**Keywords:** Bell’s palsy, surgical decompression, facial nerve

## Abstract

(1) Background: The effectiveness of decompression surgery for Bell’s palsy is controversial. This study investigated the effects of facial nerve decompression in patients with severe Bell’s palsy who were expected to have a poor prognosis. (2) Methods: We retrospectively reviewed 1721 patients with Bell’s palsy who visited the Kyung Hee University Hospital between January 2005 and December 2021. Of these, 45 patients with severe Bell’s palsy were divided into two groups; 30 patients were treated conservatively with steroids and antiviral agents alone, while 15 patients underwent additional decompressive surgery after the conservative treatment. Outcomes were measured using House–Brackmann (H–B) grade for least 6 months after treatment was finished and conducted until full recovery was achieved. (3) Results: There was no significant difference in the rate of favorable recovery (H–B grade 1 or 2) between the surgery group and the conservative treatment group (75% vs. 70.0%, *p* > 0.05). Although H–B grade improvement occurred in both groups, the degree of improvement was not significantly different between groups. (4) Conclusions: Facial nerve decompression surgery in severe Bell’s palsy patients did not significantly improve prognosis beyond that offered by conservative treatment alone. Additional surgical decompression may not be necessary in patients with severe Bell’s palsy if they receive sufficient conservative treatment.

## 1. Introduction

Bell’s palsy is an acute, unilateral, peripheral facial nerve paresis or paralysis of unknown cause. It has an annual incidence of 20–30 per 100,000 population [1,2]. The pathophysiology of Bell’s palsy includes edematous swelling of the facial nerve within the Fallopian canal, which causes a conduction block and subsequent dysfunction. However, the exact mechanism of impaired nerve function remains unclear. Although most patients with Bell’s palsy have a good prognosis, approximately 10–29% of affected patients develop persistent facial nerve dysfunction [3,4]. Facial nerve lesions determine the shape of the face, which can have a significant impact on an individual’s social life. It is important to identify and treat patients at high risk of poor long-term outcomes in order to reduce the risk of persistent facial nerve dysfunction and psychological distress [5].

There is strong evidence supporting the use of steroids as the initial medical treatment in Bell’s palsy to reduce facial nerve inflammation [6,7]. Treatment guidelines for Bell’s palsy published by the American Academy of Otolaryngology concluded that treatment with oral steroids within 72 h of symptom onset is highly likely to be effective in new-onset Bell’s palsy patients with or without concurrent antiviral therapy [8]. However, some patients do not respond to conservative treatment, and suffer sequelae, including facial asymmetry, contracture, and synkinesis [9].

Electroneuronography (ENoG) and electromyography (EMG) are objective electrical tests that are used to estimate the degree of paralysis and prognosis in Bell’s palsy patients. Patients with Bell’s palsy are at a high risk of poor recovery if they demonstrate the following clinical findings: a complete lack of facial movement on clinical examination; ENoG findings of >90% degeneration; and EMG findings of no voluntary motor unit potentials [3,10,11,12]. 

For patients at risk of poor recovery, surgical decompression of the facial nerve has been proposed as an additional treatment option to release nerve entrapment in the facial canal and improve outcomes [13,14,15]. However, to date, there is no consensus on surgical decompression. There are various opinions regarding the optimal timing of surgery and surgical approach. Some authors have argued that the most favorable outcomes are obtained when decompression surgery is performed within 14 days after symptom onset [14,16]. In contrast, others suggest that the benefits from delayed decompression surgery occur between 3 weeks and 4 months after symptom onset [13,17]. With regard to the surgical approach, some authors have proposed that the middle fossa approach offers good access to the labyrinthine segment of the facial nerve with an acceptable complication risk [16]. However, other studies suggested that the transmastoid approach is effective enough to decompress the facial nerve with a relatively low complication rate [13].

The purpose of this study was to investigate the effects of surgical decompression in patients with severe Bell’s palsy who were expected to have a poor prognosis based on clinical and electrodiagnostic tests. This study also uses the House–Brackmann (H–B) grade to compare the outcomes of decompression surgery with those of conservative treatment alone.

## 2. Materials and Methods

We conducted a retrospective case review of 1721 patients with Bell’s palsy who visited the outpatient clinic in the Department of Otolaryngology at Kyung Hee University Hospital between January 2005 and December 2018. A total of 775 patients, who were treated with high-dose steroids and antiviral agents within one week of diagnosis, were enrolled. The steroid treatment schedule consisted of prednisolone 80 mg/day for the first 4 days, 60 mg/day for 2 days, 40 mg/day for 2 days, 20 mg/day for 2 days, and 10 mg/day for 2 days. Patients were also treated with antiviral agents, including 1000–2400 mg/day acyclovir for 5 days, or 750 mg/day famciclovir for 7 days.

Blood samples were obtained from all of the patients within 5 days after the onset of facial paralysis. Complete blood count (CBC), differential count (DC), and HbA1c were measured. The neutrophil to lymphocyte ratio (NLR) was defined as the absolute neutrophil count divided by the absolute lymphocyte count. The platelet-to-lymphocyte ratio (PLR) was defined as the platelet count divided by the absolute lymphocyte count. The upper limit of NLR for adults was set to 3.53 based on previous studies [18,19]. The degree of facial palsy was measured using the H–B grade [20].

Out of the 775 patients with Bell’s palsy treated with high-dose steroids and antiviral agents, 45 met the following inclusion criteria: (1) adult >16 years of age; (2) severe facial palsy on early examination (based on an H–B grade of 5 or 6); (3) degree of denervation >90% on electroneurography (ENoG) performed between 5 and 14 days from the onset of facial weakness; (4) no voluntary electromyographic (EMG) compound motor action potentials (CMAP) performed 14 days or later after the onset of facial palsy; and (5) follow-up for at least 6 months after onset. Patients with recurrent facial palsy, traumatic facial palsy, suspected Ramsay–Hunt syndrome, previous history of otologic surgery, or those who were pregnant/breastfeeding were excluded.

Ultimately, 45 patients with severe Bell’s palsy were divided into two groups according to the treatment regimen. The conservative treatment group included 30 patients who were treated with prednisolone and antiviral agents. The surgery group included 15 patients who underwent nerve decompression surgery.

In the surgery group, transmastoid decompression was performed 21–70 days after the symptom onset of Bell’s palsy by a single experienced surgeon. The facial nerve was decompressed from the distal portion of the labyrinthine segment to the stylomastoid foramen using the transmastoid approach. The incus was temporarily removed to obtain wide access to the geniculate ganglion. The geniculate ganglion and the distal part of the labyrinthine segment were exposed by removing the supralabyrinthine cells anterior to the bony superior semicircular canal. More than 50% of the surface of the facial canal could be exposed. Finally, the incus, which had been temporarily removed, was returned to its original position.

The grade of facial palsy was followed using the H–B grade for at least 6 months until full recovery was achieved. A favorable recovery was defined by a final H–B grade of 1 or 2. An H–B grade ≥3 was defined as an unfavorable recovery. The study protocol was approved by the Institutional Review Board of Kyung Hee University Hospital. Informed consent was not required, as this was retrospective study (IRB No 2019-07-065).

### Statistical Analysis

All statistical analyses were performed using IBM SPSS version 20.0 (IBM Corp., Armonk, NY, USA). Categorical variables were analyzed using the chi-square test and Fisher’s exact test. Continuous variables were compared using the independent t-test or Mann–Whitney U-test, as appropriate. Univariate analysis was performed to assess whether the treatment methods and comorbidities influenced the outcome in severe Bell’s palsy patients. Statistical significance was considered when the *p*-value was < 0.05. All values were presented as means ± standard deviations, or as frequencies and percentages.

## 3. Results

A total of 775 patients were treated with high-dose steroids and antiviral agents within 1 week of diagnosis. There were 361 men and 414 women with a mean age of 47.8 ± 16.0 years. The mean BMI was 23.9 ± 3.9. These 775 patients were classified by the severity of their initial facial paralysis, as follows: 578 (74.9%) had mild to moderate palsy (H–B grade 2, 3, 4), and 197 (25.1%) had severe palsy (H–B grade 5, 6). These two groups differed significantly in the final H–B grade (1.6 ± 0.7 vs. 2.1 ± 1.1, *p* < 0.001), ENoG score (59.3 ± 23.2 vs. 33.1 ± 25.3, *p* < 0.001), white blood cell (WBC) count (7453.6 ± 2384.7/mL vs. 8858.7 ± 3317.5/mL, *p* = 0.001), neutrophil percentage (62.6% ± 12.8% vs. 69.2% ± 12.4%, *p* = 0.001), lymphocyte percentage (28.5% ± 10.6% vs. 23.0% ± 10.1%, *p* = 0.001), NLR (2.9 ± 2.3 vs. 4.6 ± 4.3, *p* = 0.001), and the percentage with a favorable recovery (90.7% (524/54) vs. 75.8% (147/47), *p* < 0.001). In contrast, there were no differences between the groups with regard to age, BMI, or HbA1c. The prevalence of patients with hypertension and diabetes mellitus also did not differ significantly across the groups (Table 1).

Table 2 shows the demographics and clinical characteristics of severe Bell’s palsy patients (H–B grade 5, 6) with >90% denervation on ENoG and no voluntary CMAP on EMG. There were no differences in the age, sex distribution, BMI, comorbidities, final H–B grade, or all blood test variables between the groups. The rate of favorable recovery in the surgery group was higher than that in the conservative treatment group (75% (9:3) vs. 70.0% (21:9), *p* = 0.746), but the difference was not statistically significant.

We evaluated the risk factors associated with the degree of H–B grade improvement, which was defined by the difference between the initial H–B grade and the final H–B grade. We found that the H–B grade improvement degree did not differ significantly between young and old patients (3.06 vs. 2.90, *p* = 0.633), those with and without hypertension (2.88 vs. 3.12, *p* = 0.419), with and without diabetes mellitus (3.03 vs. 3.03, *p* = 0.913), with normal weight or obesity (3.13 vs. 2.88, *p* = 0.513), or with normal or high NLR (3.10 vs. 2.67, *p* = 0.375). The presence of each risk factor tended to show less improvement in the H–B grade, but this increase was not statistically significant (Figure 1).

An additional analysis was performed to compare patients who underwent conservative treatment with those who underwent surgery. Comparison of the H–B grade improvement degree revealed no significant difference between the conservative treatment group and the surgery group (2.97 vs. 3.17, *p* = 0.533) (Figure 2).

We performed univariate analysis, using age, sex, HTN, DM, BMI, and blood test variables in addition to the treatment methods. None of these factors was statistically significant in predicting unfavorable recovery in patients with severe Bell’s palsy (Table 3).

## 4. Discussion

In this study, we selected the patients with severe Bell’s palsy with poor prognosis based on their ENoG, EMG, and physical examination findings. We also evaluated the effect of surgical decompression of the facial nerve.

Previous studies have revealed that most Bell’s palsy patients are not surgical candidates, because of the excellent overall rates of recovery. Spontaneous recovery ranges from approximately 70% with no treatment, to 94% with steroids. However, long-term sequelae, including facial paresis, paralysis, and synkinesis, are still significant consequences in a small portion of Bell’s palsy patients. Electrodiagnostic testing in patients with severe paralysis can identify a subset of patients who have an increased likelihood of poor prognosis [6,21]. ENoG findings of >90% degeneration and absent volitional nerve activity on EMG have been found to predict poorer prognosis [14,22]. In this group of patients, medical treatment alone can lead to relatively poor recovery rates; therefore, more aggressive intervention could be considered [8].

In an experimental animal model, improved nerve regeneration occurred in cats with facial nerve palsy when nerve decompression was performed within 12 days of the injury [23]. The same rationale has been applied to Bell’s palsy patients because the pathophysiology is likely related to neural edema of the facial nerve inducing an intrinsic compression at its most narrow course in the bony fallopian canal. Several studies have described edematous swelling of the facial nerve in most Bell’s palsy patients for up to three months after symptom onset [13,15,17,24,25]. There are also inflammatory changes in the facial nerve in patients who underwent delayed surgical decompression [13,17]. Based on these studies, it was hypothesized that surgical decompression of the facial nerve would prevent continuous nerve degeneration due to nerve compression and therefore improve outcomes in patients at risk for persistent dysfunction.

There have been multiple studies of surgical decompression of the facial nerve in Bell’s palsy. However, there is no consensus regarding its utility in this clinical setting, especially with regard to surgical indications, the optimal timing of surgery and the appropriate surgical approach. In general, surgical decompression is performed in patients with complete paralysis after medical treatment has failed. Still, the optimal timing of surgery remains unclear.

Some studies have reported favorable outcomes after delayed surgical decompression in severe Bell’s palsy. Bodenez et al., described favorable outcomes after delayed decompression surgery (30–120 days after disease onset) in 13 patients with severe facial palsy [17]. Another study found that patients with severe Bell’s palsy could benefit from decompression surgery within 90 days of symptom onset [26]. However, these prior studies did not have control groups. Therefore, it is difficult to determine whether the beneficial outcomes were due to the decompression surgery, or to spontaneous recovery.

Although we found that surgical decompression (performed 21–70 days after symptom onset) in severe Bell’s palsy improved functional outcomes, the functional improvement was not statistically different from that of the control group at the 6-month follow-up. Contrary to our findings, Yanagihara et al., found that delayed transmastoid decompression performed 15–120 days after symptom onset may be more beneficial than steroid treatment alone in patients with severe Bell’s palsy [13]. There are several potential reasons for this difference. Although the efficacy of antiviral agents in Bell’s palsy remains unclear, the combination of steroid and antiviral agents has been suggested to be more effective than steroids alone, especially in patients with severe Bell’s palsy [27,28,29]. In our study, all patients with severe facial palsy received both oral steroids and antiviral agents for 2–3 weeks. In contrast, the control groups in most of the previous studies only received steroids. Therefore, we believe that one strength of this study is that it compares the maximum potential effect of conservative treatment with the effect of surgical treatment. Similar to our findings, a recent study found that patients with severe Bell’s palsy did not benefit from delayed decompression surgery two months after disease onset [30].

Some studies have suggested that decompression surgery results in better facial nerve outcomes if it is performed within 14 days of symptom onset than it does if performed after 14 days. Gantz et al., conducted a multi-centered, prospective, case-control study that enrolled 70 patients treated for severe Bell’s palsy. The results show that 31 of 34 (91%) patients who underwent middle fossa decompression within 14 days of disease onset achieved H–B grades 1 or 2 compared with only 15 of 36 (42%) patients treated with steroids alone [14]. These findings were supported by another retrospective study that enrolled 14 patients who underwent surgical decompression for severe Bell’s palsy within 14 days of symptom onset. At 1 year of follow-up, 10 patients (71.4%) recovered to H–B grade 1 or 2, while the remaining 4 (28.6%) recovered to grade 3 [16]. Despite these findings, performing early surgical decompression within 2 weeks of symptom onset may be subject to various clinical limitations. The clinical guideline committee, assembled by the American Academy of Otolaryngology, did not recommend surgical decompression in Bell’s palsy because of the variable quality of the evidence and because the surgical risk may outweigh the potential benefits [8]. Decompression surgery of the facial nerve has significant costs and rare, but serious risks. Therefore, it is reasonable to consider surgery after conservative treatment has been performed for a sufficient period of time with appropriate dosing, which requires approximately two weeks. The timing of EMG testing may also influence the timing of surgical treatment because muscle denervation potentials are best observed at least two weeks after symptom onset. These EMG findings include positive sharp wave or fibrillation potentials. Decompression surgery should only be recommended if conservative treatment is ineffective and the prognosis is expected to be worse with electrophysiologic testing. Therefore, this decision should only be considered at least two weeks after symptom onset. In this study, all patients with severe Bell’s palsy underwent conservative treatment preferentially for 2–3 weeks without early decompression surgery. Surgical decompression was considered after three weeks of symptom onset.

The optimal approach of decompression surgery for Bell’s palsy is still debated. Several studies recommend the middle fossa approach to reach the bottle neck of the fallopian canal from the internal auditory canal [14]. One previous study concluded that facial nerve decompression should be performed using the middle fossa approach within two weeks of symptom onset and it reported a 22.2% complication rate of only temporary minor complications, including facial numbness/tingling, nausea/vomiting, dizziness, periorbital ecchymosis, and pruritus [16]. However, the use of the middle fossa approach has been associated with rare but serious risks, including hearing loss, cerebrospinal fluid leak, hematoma formation, seizure, aphasia, and stroke. These more serious side effects are due to the fact that the middle fossa approach requires a craniotomy and elevation of the temporal lobe [3,31,32]. This approach is not typically recommended for patients 65 years or older because of the fragile nature of their dura [12]. Therefore, based on their clinical experience, the transmastoid approach has been accepted by some surgeons as an option for facial nerve decompression. Several studies have reported improved outcomes with the transmastoid approach compared to those of conservative treatment alone [13,17,26]. In this study, all patients in the surgery group underwent the transmastoid surgical decompression considering the treatment effects and side effects reported in previous studies.

Although the use of surgical facial nerve decompression is still a topic of debate. it is used in patients of complete paralysis with evidence of extensive neurodegeneration who, on nerve conduction test, show conduction velocities on the lesion side that are less than 10% of those on the unaffected side. When the voluntary motor unit potential is not observed in the needle electromyography test, a therapeutic effect can be expected if the surgical facial neve decompression is performed early.

## 5. Conclusions

Patients who underwent decompression surgery 21–70 days after symptom onset of severe Bell’s palsy did not show better prognosis than did those who were treated with conservative treatment alone. These findings are consistent with the previous guideline and systematic reviews, which have not recommended surgical decompression for patients with Bell’s palsy [33]. Our results suggest that patients with severe Bell’s palsy do not benefit from the decompression surgery. Therefore, additional surgical decompression of Bell’s palsy may not be necessary if sufficient conservative treatment is provided.

## Figures and Tables

**Figure 1 jcm-11-00888-f001:**
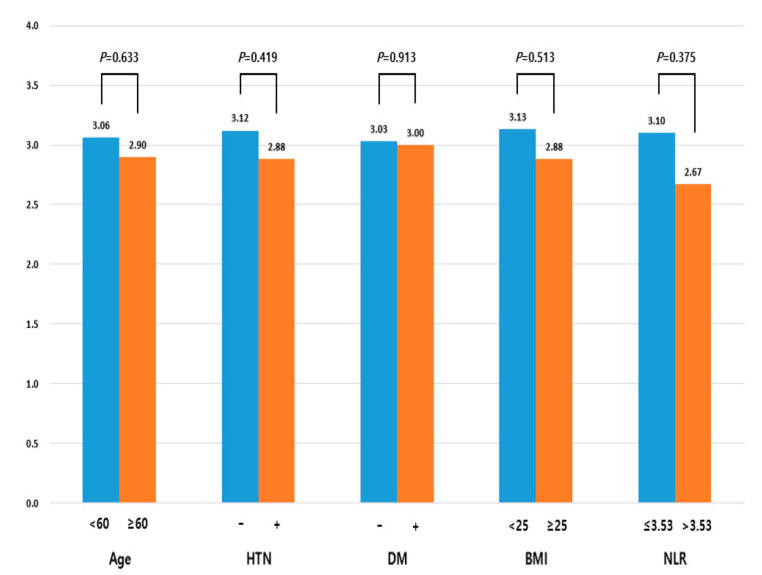
Relationships between risk factors and degree of H–B grade improvement. degree of H–B grade improvement—initial H–B grade–final H–B grade.

**Figure 2 jcm-11-00888-f002:**
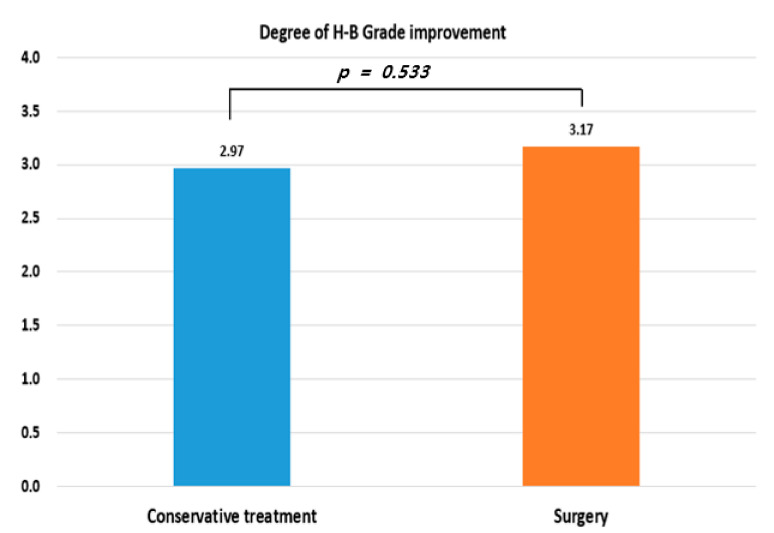
H–B grade improvement by group.

**Table 1 jcm-11-00888-t001:** Baseline demographics and clinical characteristics of patients with Bell’s palsy.

	Total	Mild-to-Moderate Palsy	Severe Palsy	*p*-Value
no.	775	578	197	
age	47.8 ± 16.0	48.1 ± 15.7	46.7 ± 16.9	0.292
sex (male:female)	361:414	273:305	88:109	0.483
BMI (kg/m^2^)	23.9 ± 3.9	23.8 ± 4.1	24.0 ± 3.6	0.786
comorbidity				
HTN: none-	252:520	194:384	58:139	0.297
DM: none-	117:655	85:493	32:165	0.477
H–B grade				
initial	3.6 ± 1.1	3.1 ± 0.8	5.0 ± 0.2	<0.001 *
final	1.7 ± 0.8	1.6 ± 0.7	2.1 ± 1.1	<0.001 *
ENoG (%)	47.9 ± 27.4	59.3 ± 23.2	33.1 ± 25.3	<0.001 *
blood test				
HbA1c (%)	5.9 ± 0.9	6.0 ± 1.0	5.9 ± 0.8	0.400
WBC (/mL)	8026.3 ± 2878.5	7453.6 ± 2384.7	8858.7 ± 3317.5	0.001 *
platelet (/mL)	273,092.4 ± 81,349.5	267,862.4 ± 72,359.4	280,693.3 ± 92,871.3	0.294
monocyte (%)	4.7 ± 1.9	4.8 ± 1.9	4.5 ± 1.8	0.215
eosinophil (%)	1.6 ± 1.6	1.8 ± 1.7	1.4 ± 1.5	0.172
basophil (%)	0.4 ± 0.3	0.4 ± 0.2	0.4 ± 0.3	0.761
neutrophil (%)	65.3 ± 13.0	62.6 ± 12.8	69.2 ± 12.4	0.001 *
lymphocyte (%)	26.2 ± 10.7	28.5 ± 10.6	23.0 ± 10.1	0.001 *
NLR	3.6 ± 3.4	2.9 ± 2.3	4.6 ± 4.3	0.001 *
PLR	166.2 ± 96.2	155.8 ± 84.5	181.2 ± 109.9	0.079
Recovery ^(a)^(favorable: unfavorable)	671:101	524:54	148:49	<0.001 *

Values are presented as means ± standard deviations. BMI—body mass index; HTN—hypertension; DM—diabetes mellitus; H–B grade—House–Brackmann grade; ENoG—electroneuronography; HbA1c—glycated hemoglobin; WBC—white blood cell; NLR—neutrophil to lymphocyte ratio; PLR—platelet to lymphocyte ratio; mild-to-moderate palsy—initial H–B grades 2, 3, 4; severe palsy—initial H–B grades 5, 6; ^(a)^ favorable recovery—final H–B grades 1, 2; unfavorable recovery—final H–B grades 3, 4, 5, 6, * *p* < 0.05.

**Table 2 jcm-11-00888-t002:** Demographics and clinical characteristics of patients with severe Bell’s palsy ^(a)^.

	Conservative Treatment ^(b)^	Surgery ^(c)^	*p*-Value
no.	30	15	
age	49.6 ± 15.0	48.5 ± 17.4	0.838
sex (male:female)	14:16	7:8	0.769
BMI (kg/m^2)^	24.8 ± 3.3	23.9 ± 3.5	0.594
comorbidity			
HTN: none-	13:17	3:12	0.316
DM: none-	9:21	4:11	0.833
H–B Grade			
Initial	5.1 ± 0.3	5.2 ± 0.4	0.558
final	2.1 ± 1.1	2.0 ± 0.7	0.703
blood test			
HbA1c (%)	7.0 ± 1.5	6.1 ± 0.7	0.179
WBC (/mL)	7674.0 ± 2143.8	8991.3 ± 3349.2	0.453
platelet (/mL)	224,000.0 ± 114,435.6	242,125.0 ± 45,256.2	0.691
monocyte (%)	4.5 ± 0.2	5.5 ± 1.4	0.181
eosinophil (%)	1.3 ± 1.0	1.1 ± 0.8	0.764
basophil (%)	0.3 ± 0.1	0.7 ± 0.7	0.268
neutrophil (%)	64.1 ± 13.1	67.1 ± 9.6	0.640
lymphocyte (%)	28.4 ± 11.8	23.7 ± 8.8	0.421
NLR	2.9 ± 1.9	4.2 ± 4.6	0.553
PLR	100.2 ± 56.4	133.8 ± 28.7	0.178
Recovery(favorable: unfavorable)	21:9	10:5	0.746

Values are presented as means ± standard deviations. BMI—body mass index; HTN—hypertension; DM—diabetes mellitus; H–B grade—House–Brackmann grade; HbA1c—glycated hemoglobin; WBC—white blood cell; NLR—neutrophil to lymphocyte ratio; PLR—platelet to lymphocyte ratio; ^(a)^ severe palsy—initial H–B grades 5, 6; ^(b)^ conservative treatment—steroid + antiviral agent; ^(c)^ surgery—transmastoid decompression surgery

**Table 3 jcm-11-00888-t003:** Univariate analysis for unfavorable recovery ^(^^a)^ in patients with severe Bell’s palsy.

	Odds Ratio (95% Confidence Interval)	*p*-Value
conservative treatment only	0.33 (0.28–5.89)	0.099
age < 60 years	0.43 (0.19–4.33)	0.220
female	0.58 (0.12–1.86)	0.257
HTN	2.00 (0.51–7.81)	0.319
DM	1.17 (0.28–4.88)	0.833
BMI	0.97 (0.71–1.33)	0.853
neutrophil	1.02 (0.91–1.15)	0.711
lymphocyte	0.97 (0.85–1.10)	0.595
monocyte	2.90 (0.78–10.87)	0.113
eosinophil	1.46 (0.36–5.82)	0.595
WBC	1.00 (0.99–1.00)	0.213
platelet	1.00 (1.00–1.00)	0.201
NLR	0.95 (0.66–1.37)	0.786
PLR	0.98 (0.95–1.01)	0.244

^(a)^ Unfavorable recovery: Final H–B grades 3, 4, 5, 6.

## Data Availability

The data presented in this study are available on request from the corresponding author. The data are not publicly available because the health examination data of a private hospital was used.

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
