# Peer review of "Comparison of Medical and Surgical Treatment in Severe Bell’s Palsy"

_jcm, 2022, doi:10.3390/jcm11030888_

Round 1

Reviewer 1 Report

The authors have provided a retrospective review of 1721 patients who presented at Kyung Hee University Hospital between 2005-2021 with symptoms of Bell’s palsy.  They have provided demographic information of 775 patients treated with high-dose steroids and antiviral agents within a week of diagnosis.  They have further sub-selected 45 adult patients who were diagnosed with severe Bell’s palsy, based on House-Brackmann (H-B) grade, electroneurography score, electromyographic compound motor action potentials and record of follow-up for at least 6 months after onset.  They report on the outcomes of these 45 patients with severe Bell’s palsy who were treated either conservatively with steroid and antiviral medications, or with medications and transmastoid decompression 21-70 days after symptom onset.  Recorded H-B grades were followed for at least 6 months until full recovery.

The manuscript is well crafted.  There are a few components of the manuscript that could be clarified for readers.

  • The methods section indicates there were 772 patients included in the larger study (page 2, line 76) or 775 patients (page 2, line 89). Results are provided for 775 patients (page 3, line 125 and Table 1).
  • Review of patient records was carried out for at least 6 months or to the point of “full recovery”, defined as H-B grade of 1 or 2. Some patients did not achieve “full recovery”.  It is assumed that the time point for the unfavorable recovery (H-B grade equal to or greater than 3) was 6 months, however this is not stated clearly in the methods (page 3, lines 111-113).
  • In the results section, it is not easy to determine which of the data were obtained at baseline and which were obtained at the final point of follow-up or recovery. For example, in the text, “final” scores are provided for H-B grade, blood cell counts, and the percentage of patients with favorable recovery based on H-B grade (page 3, lines 130-135).  However, Table 1 is entitled, “Baseline demographics and clinical characteristics”. Perhaps the table could clearly indicate which data were obtained early in the diagnosis (are demographic) and which were obtained at the point of full recovery or the 6-month cut-off.  Alternatively, the table legend could state clearly that ENoG and blood counts were obtained at the point of recovery.
  • Clarity in timing of the blood count data is necessary for interpretation of the univariate analysis, as blood counts are factored into the analysis, along with comorbidities (page 4, lines 163-165, and Table 3). This analysis would suggest that blood counts were obtained at some early point in the disease process.  Please clarify the timing of the blood counts.
  • One statement in the discussion section would benefit from inclusion of references. “Several studies have described edematous swelling of the facial nerve in most Bell’s palsy patients for up to three months after symptom onset” (page 7, lines 198-200).
  • In the present study, all the patients who underwent surgical decompression were treated by the same physician surgeon. Given that the discussion section includes general rationale for surgical decompression, the authors may wish to include in the discussion the criteria used by the surgeon in the present study.
  • In the larger group of 775 patients, the final mean ENoG (%) was lower in patients with severe palsy, when compared with patients with mild to moderate palsy (Table 1). Was this expected?  Is there precedent for this in the extant literature?  This could be addresses in the discussion section.

Author Response

We attached the file. Thank you.

Reviewer 2 Report

The authors investigated the effects of surgical decompression in patients with severe Bell’s palsy compared to conservative treatment. This is an article offering valuable insight for additional surgical decompression in severe Bell’s palsy patients.

  • Referring to the numbers in the COMORBIDITY HTN and DM, and RECOVERY columns in Table 2, the total number of patients in the Surgery group is 12 cases. Were there 3 omission cases? The rate of favorable recovery could be changed depending on whether these 3 cases are classified into Favorable or Unfavorable. Authors need to check them again. If you do not have the data of these 3 cases, you should define Surgery group as 12 cases?
  • In Table 1, the rate of favorable recovery of Severe palsy group (197 cases) is 75.8%. In Table 2, the rate of favorable recovery of Conservative treatment group and Surgery group are 70%, 75%, respectively. I have an impression whether these therapeutic interventions are not contributing to the improvement of prognosis for severe Bell’s palsy patients. Do you have data about that of patients without these treatment in Severe palsy group. It could be informative for readers.
  • Are there any adverse events or complications of the surgical decompression in this study?
  • Line 76: 772 patients → 775 patients

Author Response

We attached the file. Thank you.
